# Transcriptomics and Metabolomics Analyses Reveal High Induction of the Phenolamide Pathway in Tomato Plants Attacked by the Leafminer *Tuta absoluta*

**DOI:** 10.3390/metabo12060484

**Published:** 2022-05-26

**Authors:** Marwa Roumani, Jacques Le Bot, Michel Boisbrun, Florent Magot, Arthur Péré, Christophe Robin, Frédérique Hilliou, Romain Larbat

**Affiliations:** 1UMR1121 Laboratoire Agronomie et Environnement (LAE), Université de Lorraine, INRAE, F-54000 Nancy, France; marwa.roumani@univ-lorraine.fr (M.R.); florent.magot@univ-lorraine.fr (F.M.); christophe.robin@univ-lorraine.fr (C.R.); 2INRAE, UR1115 Plantes et Systèmes de Culture Horticole, 84000 Avignon, France; jacques.lebot@inrae.fr; 3CNRS, L2CM, Université de Lorraine, F-54000 Nancy, France; michel.boisbrun@univ-lorraine.fr; 4INRAE, Institut Sophia Agrobiotech, Université Côte D’Azur, CNRS, F-06903 Sophia Antipolis, France; arthur.pere@inrae.fr (A.P.); frederique.hilliou@inrae.fr (F.H.)

**Keywords:** tomato, leafminer, *Tuta absoluta*, phenolamide, metabolomics, transcriptomics, herbivory

## Abstract

Tomato plants are attacked by a variety of herbivore pests and among them, the leafminer *Tuta absoluta*, which is currently a major threat to global tomato production. Although the commercial tomato is susceptible to *T. absoluta* attacks, a better understanding of the defensive plant responses to this pest will help in defining plant resistance traits and broaden the range of agronomic levers that can be used for an effective integrated pest management strategy over the crop cycle. In this study, we developed an integrative approach combining untargeted metabolomic and transcriptomic analyses to characterize the local and systemic metabolic responses of young tomato plants to *T. absoluta* larvae herbivory. From metabolomic analyses, the tomato response appeared to be both local and systemic, with a local response in infested leaves being much more intense than in other parts of the plant. The main response was a massive accumulation of phenolamides with great structural diversity, including rare derivatives composed of spermine and dihydrocinnamic acids. The accumulation of this family of specialized metabolites was supported by transcriptomic data, which showed induction of both phenylpropanoid and polyamine precursor pathways. Moreover, our transcriptomic data identified two genes strongly induced by *T. absoluta* herbivory, that we functionally characterized as putrescine hydroxycinnamoyl transferases. They catalyze the biosynthesis of several phenolamides, among which is caffeoylputrescine. Overall, this study provided new mechanistic clues of the tomato/*T. absoluta* interaction.

## 1. Introduction

The co-evolution of plants and insect herbivores over hundreds of millions of years has shaped a complex and multifaceted defense system of plants against herbivory. Plant defense strategies include morphological and chemical, constitutive and inducible, local and systemic processes, driven by sophisticated pest detection and information signaling [1,2,3,4,5]. In agronomy, understanding and taking advantage of these plant defense mechanisms are of major interest in order to develop pest control strategies alleviating or suppressing the use of pesticides.

Tomato is a major horticultural crop producing over 180 million tons of fruit worldwide [6]. Tomato plants can be attacked by a large array of herbivorous insects, including sap-feeding, chewing and leaf-mining pests. Among the latter, the South American tomato pinworm, *Tuta absoluta* (Meyrick) (Lepidoptera: Gelechiidae), is one of the main threats due to its recent and rapid expansion in the last 15 years from South America to the main world production areas. The pest can cause up to 80–100% yield losses when no control method is used [7,8,9]. Recently, there have been major advances in the management of *T. absoluta*, allowing the deployment of several biocontrol levers to limit the impact of this pest, including essential oil based-pesticides, pheromone trapping and biological control under integrated pest management (IPM) practices [7]. Despite strong fundamental advances in the understanding of tomato resistance traits against *T. absoluta*, this knowledge still needs to be improved before it can be converted into new practices in IPM strategies.

Commercial tomatoes (*Solanum lycopersicum* L.) are highly susceptible to *T. absoluta* but recent screenings on tomato populations have led to identify moderately resistant cultivars [10,11,12,13]. Some resistance traits have, however, been identified in wild tomato relatives. They correlate with the constitutive content of several leaf chemicals, namely methyl-ketones, sesquiterpenes and acyl sugars, produced by glandular trichomes, opening the door for genetic programs to introgress this resistance trait into cultivated tomatoes [14,15,16]. In addition, during *T. absoluta* infestation, tomato plants can induce defense mechanisms mediated by the accumulation of defense enzymes (polyphenol oxidase and peroxidase) at the insect feeding site as well as in distant organs [17] and also by changes in the composition of the blend of volatile organic compounds (VOCs) emitted by the infested plants [18,19,20]. Interestingly, some of these induced VOCs were found to attract natural enemies of *T. absoluta*, such as *Macrolophus pygmaeus* Rambur, *Macrolophus basicornis* Stal, *Engytatus varians* Distant, *Campyloneuropsis infumatus* Carvalho and *Nesidiocoris tenuis* Reuter, suggesting a functional indirect defense strategy [21,22,23]. In addition, very recently, the (*Z*)-3-hexenyl propanoate, a green leaf volatile commonly emitted by plant submitted to herbivory, was found to induce a defense mechanism in tomato under lab conditions, by significantly impairing *T. absoluta* infestation, via both antixenosis and antibiosis mechanisms [24]. Those authors demonstrated also the efficiency of (*Z*)-3-hexenyl propanoate application under a commercial greenhouse condition, thus defining a first proof of concept of the efficiency of the tomato defense induction strategy for the control of *T. absoluta* [25].

Since a plant’s response is usually pest specific, characterizing the response of tomato defense to *T. absoluta* herbivory seems essential to target the key metabolic pathways, metabolites or genes that are able to impact this pest. To the best of our knowledge, only one attempt was conducted through nuclear magnetic resonance (NMR) analyses with a targeted metabolomics approach following 18 metabolites. It led to identifying the accumulation of trigonellin in infested plants and highlighted a higher content of some phenylpropanoids in a tomato accession exhibiting less sensitivity to the *T. absoluta* herbivory [26].

In the present study, our main objective was to characterize the metabolic responses of young tomato plants to the herbivory of *T. absoluta* larvae. We combined untargeted metabolomic and transcriptomic analyses on several plant parts, reflecting both local and systemic responses, and at different harvest times. This multi-omics approach allowed a comprehensive analysis of the tomato response to *T. absoluta* herbivory and enabled the functional identification of tomato genes involved in plant defense.

## 2. Results

### 2.1. Metabolomics Analysis of Spatio-Temporal Responses of Tomato to Herbivory

In order to address the question of spatio-temporal induction of the tomato soluble defense against *T. absoluta* herbivory, we conducted an untargeted metabolomics study of five vegetative parts of tomato plants submitted or not to *T. absoluta* herbivory for 4 and 7 days. A total of 2427 features were detected among all the samples. A two-way analysis of variance (ANOVA) conducted on the feature ion intensities of the datasets at 4 and 7 days revealed that the organ nature impacted significantly (*p* < 0.05) around 90% of the detected features (Figure 1A; Appendix A). *T. absoluta* herbivory affected only 190 and 148 features after 4 and 7 days, respectively, which represented 7% and 6% of the total features (Figure 1A; Appendix A). The large metabolic differences between roots, stems and leaves were demonstrated by the hierarchical clustering analysis that allowed also to identify a small cluster of features positively correlated to *T. absoluta* herbivory (Figure 1B; Appendix A). The strongest response occurred in the infested leaf analyzed after 4 days of herbivory. In this organ class and after 4 days of herbivory, the differential analysis between *T. absoluta* infested and control plant showed that 331 features over-accumulated and 43 decreased compared to the controls (Figure 1C). A same partitioning of up- and down-regulated features was observed after 7 days of herbivory, but with a smaller amplitude (Figure 1C). The *T. absoluta* herbivory impacted also more distant plant parts, which are mainly characterized by an over-accumulation of features. Based on the number of features affected, the plant response appeared less marked in distant plant parts than in the infested leaf. The response was weak in the leaves below the infested leaf (Figure 1C).

We identified 76 metabolites from the cumulated 493 features highlighted from the differential analysis (Table 1; Appendix A). More than half of these identified compounds belonged to the phenolamide subfamily. They covered a high diversity of phenolamide structures with derivatives containing putrescine, agmatine, lysine and tyramine as amine moieties, but also fewer common forms consisting in spermine and spermidine chains substituted with cinnamic and dihydrocinnamic acid derivatives (Figure 2). Among them, *N*,*N*,*N*-dihydrocaffeoylspermine, whose identity was confirmed by comparison with authentic chemically synthesized standard, appeared as the most accumulated phenolamide in the infested leaves. Taking advantage of the fragmentation data from some commercial standards, we were able to annotate some glycosylated phenolamides (Table 1). All the phenolamides were significantly overaccumulated in the infested leaf after 4 days, 7 days or at both time with fold change (FC) varying from 2 (*p*-coumaroylputrescine at 7 days) to 90 (dihydrocaffeoylputrescine at 4 days). In the other plant parts, phenolamide overaccumulation was restricted to only several structures and with lower amplitude (FC below 13 in all the other plant parts, Table 1). Besides phenolamides, the other identified compounds were less structurally related each other (Figure 2). The infested leaf accumulated 5′-S-methyl-5′-thioadenosine than the control. An increase in glycoalkaloid (dehydrotomatine) was detected specifically in stems after 7 days of herbivory. In addition, some metabolites were found exclusively in infested leaves, in the presence of the larvae (putatively annotated to proclavanic acid and butyryl-carnitine). They may be regarded, therefore, either as larvae specific compounds or larvae-metabolized tomato metabolites (Table 1).

### 2.2. Transcriptomics Analyses

The transcriptomic response was determined by comparing the transcriptome of *T. absoluta* infested leaves after 5 h and 24 h of herbivory with leaves from control plants. RNA sequencing produced an average of 30 million reads for each of the 12 samples (triplicates for *T. absoluta* infested vs. control at two dates). Approximately 18,200 transcripts were mapped to the tomato reference genome (ITAG2.4) (Appendix A). Differentially expressed genes (DEGs) were identified by comparing the *T. absoluta* infested vs. control at each time. The herbivory of *T. absoluta* led to 2456 DEGs after 5 h and 4269 DEGs after 24 h. After 5 h of herbivory, the proportion of up-regulated DEGs was slightly higher than the down-regulated one. This proportion was similar after 24 h (Figure 3). Among the 1318 DEGs common to 5 h and 24 h of herbivory, two third corresponded to up-regulation. The list of the up- and down-regulated DEGs at each date is given in Appendix A.

We applied gene set enrichment analysis (GSEA) based on the gene ontology (GO) of the DEGs in order to identify which pathways are affected by *T. absoluta* herbivory. About 111 GO categories were significantly affected by herbivory (Appendix A). Since a lot of GO categories were highly redundant, a refined list was established to limit the redundancy and to highlight the metabolic processes diversity affected by herbivory (Figure 4). GSEA highlighted a clear time-dependent tomato response to the herbivory with only two GO categories being commonly over-represented in the up-regulated DEGs after 5 h and 24 h. The first one, labelled «response to wounding», includes 11 and 12 genes annotated as proteinase and chymotrypsin inhibitors. The second one, «aromatic amino acid family metabolism», gathers genes involved in the biosynthesis of phenylalanine and tyrosine (dehydroquinate synthase, *Solyc02g083590*; chorismate mutase, *Solyc02g088460*; prephenate dehydratases, *Solyc06g074530* and *Solyc11g066890*) but also 5 phenylalanine ammonia lyase (PAL) genes which catalyzes the initial step leading to the phenylpropanoid pathway (*Solyc09g007890*, *Solyc09g007900*, *Solyc09g007910*, *Solyc09g007920*, *Solyc10g086180*). 

After 5 h of herbivory, the two categories «isoprenoid biosynthetic process» and «steroid metabolic process» appeared as being over-represented in the up-regulated DEGs. This suggests an activation of the whole terpenoid pathway, starting from the first steps common to all terpenoid subclasses (mevalonate kinase, *Solyc01g098840*; 3-hydroxy-3-methylglutarylCoA reductase, *Solyc02g082260*; isopentenyl diphosphate isomerase, *Solyc05g055760*; hydroxymethylglutaryl-CoA synthase, *Solyc08g080170*; farnesyl pyrophosphate synthase, *Solyc12g015860*) to more specific steps involved in the steroid backbone formation crucial for the widely distributed sterols but also the solanaceous-specific glycoalkaloids (squalene monooxygenase, *Solyc04g077440*; sterol reductases, *Solyc06g074090* and *Solyc09g009040*). The “lipid metabolism” category appeared overrepresented in both the up- and down-regulated genes. Regarding the up-regulated genes of this category, half of them were related to the terpenoid metabolism and overlapped with the categories mentioned above. The other half contained a majority of genes associated to (i) lipid catabolism, with notably 17 genes annotated as lipase or lipase-like (*Solyc01g079600*, *Solyc02g077420*, *Solyc11g065530*, *Solyc08g006850*, etc.), (ii) lipid modification (diacylglycerol O-acyltransferase, *Solyc01g095960*; O-acyltransferase WSD1 *Solyc07g053890*, phosphatidylcholine-sterol O-acyltransferase, *Solyc09g072710*…) and (iii) one gene, *Solyc02g085730*, coding for an allene oxide cyclase involved in the jasmonic acid pathway. The down-regulated genes were mainly associated to the phosphatidic acid metabolism (phosphatidylinositol-specific phospholipase C, *Solyc01g111260*, *Solyc10g055240*; phospholipase D, *Solyc02g083340*; phosphatidylinositol-4-phosphate 5-kinase family protein, *Solyc11g013830*, *Solyc11g013180*, *Solyc06g065540*) and also to the biosynthesis of very long chain fatty acid (VLCFA) (fatty acid elongase 3-ketoacyl-CoA synthase, *Solyc03g005320*, *Solyc05g009270*, *Solyc08g067260*, *Solyc09g065780*, *Solyc10g009240*). 

The tomato plant response after 24 h of herbivory was characterized by an over-representation of genes associated to signaling in up-regulated DEGs. Indeed, five genes were associated to ethylene signaling protein or ethylene receptor (*Solyc05g055070*, *Solyc06g053710*, *Solyc09g007870*, *Solyc11g006180*, and *Solyc12g011330*), whereas the over-represented GO category «phosphorylation» regrouped a majority of receptor-like kinases (RLK) whose role in signaling during the plant response to bio-aggressors is well known [28]. In addition, numerous genes identified or annotated as mitogen-activated protein kinase (MAPK) or calcium-dependent protein kinase (CDPK) were also found in this category. This is the case for *Sl*MPK1,3,4 and 5 (*Solyc12g019460*, *Solyc06g005170*, *Solyc11g072630*, and *Solyc01g094960*), which have been shown to contribute to the tomato defense response to *Botrytis cinerea* [29]. After 24 h of herbivory, a large list of GO categories appeared linked to the down-regulated DEGs. These categories covered a large array of metabolic processes, including central metabolism, the major affected categories being linked to chloroplast metabolism (photosynthesis, tetrapyrrole metabolism and carotenoid metabolism). 

KEGG (Kyoto Encyclopedia of Genes and Genomes) enrichment pathway analysis was conducted in parallel and led to similar interpretation. Indeed, the phenylpropanoid pathway appeared as the only pathway up-regulated at 5 h and 24 h of herbivory, whereas the lipid, chlorophyll and riboflavin metabolisms were significantly over-represented in the down-regulated DEGs at 5 h and 24 h and photosynthesis specifically at 24 h (Appendix A).

In addition to the GSEA and KEGG enrichment pathway approaches, transcriptomics data were also explored in light of the metabolomics analysis, which highlights a large phenolamide accumulation in the infested tissues. Consequently, we focused our analysis specifically on pathways related to the polyamine biosynthesis and the phenylpropanoid biosynthesis. Seventeen genes were followed regarding the polyamine biosynthesis (Appendix A). Five of them were strongly induced after both 5 h and 24 h of herbivory (Figure 5). They correspond to arginases (*Solyc01g091160*, *Solyc01g091170*), arginine decarboxylase (*Solyc01g110440*), ornithine decarboxylase (*Solyc04g082030*) and spermine synthase (*Solyc03g007240*), which represents all the steps to produce agmatine and putrescine and four of the five necessary steps to produce spermine starting from arginine. Regarding the phenylpropanoid pathway, 58 genes covering the core pathway and also the flavonoid pathway were followed (Appendix A). Thirteen genes were significantly up-regulated at 5 h or 24 h but only four genes were common to the two-time conditions. At 5 h, the induced genes covered mainly the phenylpropanoid core pathway, with PAL, 4-coumarate:coenzyme A ligase (4CL), *p*-coumaroyl-CoA 3′ hydroxylase (C3′H) and hydroxycinnamoyl transferase (HCT)-like genes. At 24 h, the induced genes were 3 HCT-like and also 3 tyramine-hydroxycinnamoyl-transferase (THT) specifically involved in the biosynthesis of tyramine-containing phenolamides (Figure 5). Interestingly, all the phenylpropanoid-related genes were only slightly induced with the exception of the two HCT-like genes *Solyc11g071470* and especially *Solyc11g071480* which was the most induced gene at 5 h. Despite their annotation, Blast analysis of *Solyc11g071470* and *Solyc11g071480* revealed their high homology with AT1 a putrescine hydroxycinnamoyl transferase (PHT) functionally identified in *Nicotiana attenuata* [30]. To confirm the nature of these two genes, we cloned them, heterologously expressed the protein in bacteria and assayed the purified proteins for PHT activity. The assays confirmed that Solyc11g071470 and Solyc11g071480 both exhibited the ability to synthesize caffeoylputrescine from caffeoyl-CoA and putrescine as a main activity (Figure 6). Both enzymes were also able to synthesize feruloylputrescine, caffeoylagmatine and dihydrocaffeoylputrescine but with a lower efficacity. These enzymes were, however, unable to produce spermidine- nor spermine-based phenolamides.

## 3. Discussion

The combination of transcriptomics and metabolomics analyses, taken together with the study at different harvest times and on different plant parts, allowed to obtain an exhaustive picture of the tomato plant responses to *T. absoluta* herbivory. Based on the metabolomics data, these responses were both local and systemic, the local response in infested leaves being much more intense than in other plant parts. In addition, systemic responses were mostly a subset of the metabolites induced locally with the exception of the roots whose metabolome appeared clearly to be different from that of the above-ground parts. This feature of the tomato response to the leafminer *T. absoluta* is thus shared with that against chewing herbivores, such as *Manduca sexta* [31]. Both responses promote also a massive induction of antinutritive proteinase inhibitors [32].

The tomato response to *T. absoluta* herbivory was characterized by a large and transient modulation of the terpene and lipid metabolic pathways. The induction of a wide array of genes involved in the terpenoid backbone biosynthesis, together with some terpene synthase-annotated genes and *Sl*MYC1 (*Solyc08g076930*) involved in the type VI trichome biosynthesis and terpene emission [33], suggest an induction of the synthesis/emission of volatile organic compounds (VOCs) in response to herbivory. In addition, the massive induction of lipase-annotated genes may illustrate the activation of the green leaf volatiles (GLVs) biosynthesis, initiated through the lipase-mediated degradation of galactolipids to release linoleic and α-linolenic acids. Although the emission of VOCs was not monitored in our study, this interpretation is coherent with many recent studies [18,19,20,34,35] reporting enhanced emissions of tomato VOCs in response to *T. absoluta* attacks. Moreover, the lipase-associated production of linoleic acid together with the induction of the allene oxide cyclase may suggest the induction of the jasmonic acid biosynthesis. 

Another set of genes associated to the lipid metabolism was, however, largely down regulated. They corresponded, for one part, to genes associated to the biosynthesis of phosphatidic acid (phospholipases C and D) and, for the other part, to genes mainly associated to fatty acid elongases, which are involved in the biosynthesis of cuticle and waxes. Phosphatidic acid signaling contributes to the rapid systemic plant response against biotic and abiotic stresses [36]. It is tempting to connect this apparent PA down-regulation to the low systemic response highlighted by the metabolomic analyses in rice [37]. Indeed, this recent study compared the systemic response to herbivores with different feeding guilds and identified a low systemic induction of secondary metabolites that was accompanied by a decrease in the PA associated lipid in spider mite-fed plants. Waxes and cuticle represent physical barriers that may affect plant digestive ability and also the mobility of the insects [38,39]. Overall, the apparent down-regulation of these two components of the plant defense could be the consequence of the leafminer ability to reprogram its host physiology. Such manipulations of the host plant physiology have been well described for galls and are known for leafminers [40]. To the best of our knowledge, the tomato/*T. absoluta* model has not been studied on this aspect; however, investigation on this question may be very helpful in understanding the determinants of the tomato susceptibility to this pest. 

At the metabolite level, one striking consequence of *T. absoluta* herbivory was the massive accumulation of phenolamides in tomato tissues. Although phenolamide accumulation has already been described in Solanaceae and Poaceae in response to chewing and sucking insects [30,41,42,43], it has not been reported before in this specific pathosystem [26,44,45], highlighting the potential of untargeted High Resolution Mass Spectrometry (HRMS) analyses in this kind of study. A characteristic of this phenolamine accumulation was the structural diversity of the metabolites composing this family. Besides caffeoylputrescine found in healthy tomato tissues [46], as well as tyramine, dopamine or spermidine derivatives, which are classically accumulated in attacked tomato or other solanaceous species [42,47,48,49], our analyses identified more atypical metabolites, such as spermine and spermidine derivatives associated to dihydroxycinnamoyl moities. These compounds, usually referred to as kukoamines, have been mostly reported in different plant parts of *Lycium* sp. [50,51] but also in potato tubers and tomato fruits [52]. To the best of our knowledge, these compounds have never been associated in the plant response to herbivory.

The accumulation of phenolamides was supported by transcriptomic data, which highlighted the induction of the phenylpropanoid and the polyamine pathways, providing, respectively, the (di)hydroxycinnamoyl and the amine moieties (Figure 7). Interestingly, the phenylpropanoid pathway induction was much lighter than that of the polyamine, suggesting that the latter could be rate limiting for phenolamide accumulation. A key step for the phenolamide accumulation is the association of the (di)hydroxycinnamoyl and amine moieties that is mostly catalyzed by *N*-hydroxycinnamoyltransferases [53]. *Solyc11g071470* and *Solyc11g071480*, which were highly overexpressed in response to the *T. absoluta* herbivory, were functionally characterized as putrescine hydroxycinnamoyl transferases. They complete the collection of PHT initially identified in *Nicotiana attenuata* (one gene AT1; [30]) and in *Oryza sativa* (three genes *Os04g0664600*, *Os09g0544000*, *Os09g0543900*; [54]). The genes *Solyc11g071470* and *Solyc11g071480* had comparable specificity with caffeoyl-CoA and putrescine as preferred acyl donor and acceptor, but also substantial activity with feruloyl-CoA, dihydrocaffeoyl-CoA and agmatine. Regarding their position in the tomato genome and their high nucleotide sequence identity, these two genes were probably duplicates as observed for two rice genes, *Os09g0544000* and *Os09g0543900*, exhibiting differences in their substrate preference and their expression level in different plant organs. This indicates that the two duplicates have evolved in different ways (neo-functionalization) [54]. However, *Solyc11g071470* and *Solyc11g071480* exhibited no significant difference in their substrate preference nor their stress induction. Complementary experiments on the spatial expression of these two genes in the tomato plant may help understanding the role of each gene.

The functional identification of *Solyc11g071470* and *Solyc11g071480* together with their high induction upon herbivory provide chemical support for their role in the accumulation of putrescine- and agmatine-related phenolamides (Figure 8). The question remains opened regarding the accumulation of derivatives composed by spermine/spermidine and dihydroxycinnamoyl moieties. A specific research in the transcriptomic data of tomato orthologs of reported spermidine and spermine transferases [30,55,56,57,58,59,60] followed by the functional characterization of candidates did not allow us yet to identify this crucial step (not shown).

Even if this was not the scope of this study, the question remains as to whether or not this phenolamide accumulation is a resistance trait of tomato plants to the *T. absoluta* herbivory. On the one hand, several phenolamides have been shown to have a negative impact on larval development [42], to increase insect mortality [43] or to deter insect oviposition [61]. On the other hand, other examples showed that phenolamides have no detectable impact on some insect pests [43] or were beneficial to the larval development probably because of the nitrogen feed during their digestion and assimilation [62]. Like almost all domesticated varieties, the tomato plants used for this study (cv. Better Bush) were susceptible to the herbivory of *T. absoluta*, which indicates that the phenolamide accumulation was not sufficient enough to prevent the deleterious impact of this pest. Interesting questions remain open regarding how this specialist pest bypasses the plant defense traits. Focusing on the phenolamide accumulation, the identification of genes specifically involved in their biosyntheses offers a promise to modulate the phenolamide composition and concentration in tomato lines and assess their control effects on the insects.

## 4. Materials and Methods

### 4.1. Plant and Insect Materials

Tomato seeds (*Solanum lycopersicum* L.) of the early and dwarf cultivar “Better Bush” (VFN Hybrid, breeder: Tomato Growers) were sown and grown using a hydroponic system under controlled conditions. 

*T. absoluta* adults, provided by INRAE-ISA Sophia-Antipolis (France) were reared on tomato plants grown under insectarium conditions in climatic chambers. After mating, the females laid eggs on leaves and larvae hatched out. They were selected at stage 3 (scale 1–4) to infest young tomato plants.

### 4.2. Conditions for Plant Growth

Eleven days after sowing, the seedlings were transplanted into a Nutrient Film Technique system set in a growth chamber (16 h photoperiod, 23 °C/18 °C day/night in air and 23 °C in nutrient solutions, 60% air humidity). Before the experiment, all temperature sensors (air and solution) were calibrated in a water bath with 0.01 °C precision (Haake model C35/F6, Karlsruhe, Germany).

The plants received a full nutrient solution supplying N-NO_3_^−^ at a concentration of 1 mM, which was found to be non-limiting for tomato growth [63]. Nitrogen concentration was continuously (i.e., every hour) regulated at its fixed set-points using the automated Totomatix setup [64]. The pH was monitored and regulated by Totomatix to 5.5. The detailed composition of the nutrient solution is given in Appendix A.

### 4.3. Plant Infestation by T. absoluta Larvae, Harvest and Sample Preparation

For the metabolomics analyses, 20 plants were grown. Half of them were infested 14 days after transplantation with 12 larvae laid on the three terminal leaflets of the 4th true leaf (Figure 8). The other half constituted the group of control plants, which were subjected to mock depositions of larvae, performed with a paintbrush. The plants were sampled 4 days and 7 days after larvae deposit. Leaves, stems and roots were separated (Figure 8). On the infested leaf, only the three terminal leaflets were sampled after removing the larvae. Roots were rinsed with deionized water. Organ fresh weights (FW, 0.1 mg precision) were measured, and samples were frozen in liquid N_2_ and stored at −80 °C until freeze-drying. Dry weights (DW, 0.1 mg precision) were determined, and the samples were ground to a fine powder and stored at −20 °C.

For the transcriptomics analyses, 6 plants were infested with 6 larvae deposited on the three terminal leaflets of the 4th true leaf and 6 other plants were subject to mock deposition. Two harvests were conducted after 5 h and 24 h of herbivory. At both harvest time, the infested leaflets from which the larvae had been removed (and the corresponding leaflets in the control plants) were sampled, immediately frozen in liquid nitrogen and stored at −80 °C until RNA extraction.

### 4.4. Extraction of Metabolites from Tomato Organs

Metabolites were extracted from 30 mg dry powder of leaves, stems and roots as described in [65]. The dry powder was extracted in 1 mL 60% aqueous methanol to which 50 μL of taxifolin were added (internal standard, 2 mg·mL^−1^ methanol), then blended (1 min) and centrifuged (10 min, 2800× *g*). The extraction was repeated, and the supernatants pooled and vacuum-dried. The residue was dissolved in 500 µL of 70% aqueous methanol, filtered (0.22 μm) and transferred to vials before MS^1^ analyses. One additional vial was prepared with a pool of each sample (2 µL each) for the MS^2^ analyses.

### 4.5. UHPLC-ESI-HRMS Analysis

Chromatographic analyses were performed on a Vanquish UHPLC system equipped with a binary pump, an autosampler and a temperature-controlled column. Metabolites contained in the extracts (10 µL) were separated on a XB-C18 Kinetex (150 × 2.1 mm, 2.6 µm) (Phenomenex Inc., Torrance, CA, USA) using a gradient of mobile phase composed of water + 0.1% formic acid (A) and methanol + 0.1% formic acid (B) at a flow rate of 200 μL·min^−1^. The elution program consisted in starting with 10% B for 2 min, then linearly increasing from 10 to 30% B in 8 min, then to 95% B in 10 min. The column was rinsed for 5 min with 95% B and re-equilibrated to the initial conditions for 4 min prior to the next run. The samples were analyzed randomly.

HRMS^1^ detection was performed on an Orbitrap IDX^TM^ (ThermoFisher Scientific, Bremen, Germany) mass spectrometer in positive and negative electrospray ionization (ESI) modes. The capillary voltages were set at 3.5 kV and 2.5 kV for positive and negative modes, respectively. The source gases were set (in arbitrary unit min^−1^) to 40 (sheath gas), 8 (auxiliary gas) and 1 (sweep gas) and the vaporizer temperature was 320 °C. Full scan MS^1^ spectra were acquired from 120 to 1200 *m*/*z* at a resolution of 60,000. MS^2^ analysis was performed on the pooled-samples vial using the data dependent acquisition (DDA) mode. For this analysis, the AcquireX data acquisition workflow developed by ThermoFisher was applied. Briefly, this workflow increases the number of MS^2^ acquisition, especially on low-intensity ions, through the creation of an inclusion list after a first injection of the sample and the establishment of a dynamic exclusion list occurring by the iterative sample analysis (involving 5–6 successive injections).

### 4.6. Metabolomics Data Processing

The raw UHPLC-HRMS files were uploaded on the Compound Discoverer 3.3 software (ThermoFisher Scientific, Bremen, Germany) for metabolomics analysis. The detailed analytic workflow is given in Appendix A. Briefly, it includes peak detection, chromatogram alignment and peak grouping in features. Each feature corresponds to a specific *m*/*z* at a given retention time. Compounds were identified through (i) elemental composition prediction, (ii) search in mass/formula databases, including local database and 11 public databases of which Chemspider, FoodDB, HMDB, LipidMaps and (iii) with the MS^2^ information, search in homemade and public spectral databases of which mzCloud, Mona, GNPS. When possible, the identification of metabolites was confirmed by comparison with authentic commercial standards. This was notably the case for rutin, caffeoyl-, feruloyl- and coumaroyl- putrescine and agmatine. The nature of *N*,*N*,*N*-tri-dihydrocaffeoylspermine was confirmed by comparison to the chemically synthesized compound. In a first attempt, we tried to prepare it as reported by [66] using solid-phase synthesis. Despite careful following of the procedure, in our hands the final compound was always contaminated by side-products, exhibiting more than three dihydrocaffeoyl moieties as shown by LC-HRMS. Alternatively, we prepared it by classical liquid phase synthesis. Briefly, we first protected the catechol moiety of dihydrocaffeic acid as an acetonide by standard conditions [67]. As reported by [66], we protected both ends of spermine with Dde protecting group, then one of both secondary amine group was selectively Boc-protected. Removal of Dde groups let both primary amino groups and one secondary amine free. Coupling with three equivalents of cathecol-protected dihydrocaffeic acid gave the target compound as a protected form. Final acidic treatment afforded the desired compound whose identity and purity were proven by ^1^H and ^13^C NMR spectroscopy and HRMS. Complete protocol will be given elsewhere.

Multivariate statistics on the metabolomics data were realized using the metaboanalyst platform (https://www.metaboanalyst.ca/ accessed on 20 October 2021). A two-way analysis of variance (ANOVA, *p* < 0.05) was performed to identify metabolic features affected by herbivory and organ at each sampling date. In each case, a false discovery rate approach was performed as multiple testing correction. Features differentially affected by herbivory were determined by pair comparison considering the organ nature and the harvest date by using the differential analysis application of the Compound Discoverer 3.3 software. The *p*-value of per group comparison was calculated by a one-way ANOVA, followed by the Tukey post-hoc test. 

### 4.7. Extraction of Tomato Leaf RNA

Total RNA extraction was realized from 100 mg frozen tissue (leaf, stem and root) using the E.Z.N.A. Plant RNA kit (Omega bio-tek, Norcross, GA, USA). An on-column-DNAse treatment (Qiagen, Hilden, Germany) was included during the RNA extraction. RNA quality was checked by electrophoresis on a 1% agarose gel. RNA samples were quantified using absorbance at 260 nm, their purity was evaluated by the 260/280 nm absorbance ratio and their integrity by determining and considering an RNA Integrity Number (RIN) > 7 (Agilent BioAnalyzer 2100).

### 4.8. Construction and Analysis of RNAseq Libraries

A total of 12 RNAseq libraries were constructed, corresponding to two infestation treatments (*T. absoluta* infested leaf vs. control leaf) and two harvest times (5 h and 24 h after larval deposit) in triplicates. Following the manufacturer recommendations (Eurofins Genomics, Ebersberg, Germany), 3 µg of RNA per sample were used to generate the libraries which were sequenced on an Illumina Hiseq 2500 platform to generate paired end 125 bp reads. Reads were processed and filtered using the Trimmomatic software [68]. They were mapped on the tomato genome (ITAG2.4) using the BWA-MEM software [69]. Fragments per kilobase of transcript per million mapped reads (FPKM) were calculated using the FeatureCounts software (Version 1.6.4) [70]. Data were cleaned (only transcripts > 1 cpm on at least 3 samples were selected), normalized using the TMM (Trimmed Mean of M-values) procedure implemented in the Bioconductor edgeR (v3.32.1) package [71]. The differential gene expression was conducted with the glmfit option within the edgeR package with the following parameters |log Fold Change (FC)| > 1.5 with *p*-adj value of FDR method < 0.05. Functional annotations, such as gene ontologies and interproscan features, were extracted from ITAG2.4_protein_functional.gff3 file with additional annotations on phenylpropanoid and polyamine pathways from Solgenomics (https://solgenomics.net/ accessed on 12 September 2021). Gene set enrichment analysis (GSEA) was realized using the GOfuncR (v1.10.0) [72] thanks to the gene ontology (GO) data [73,74] with GO graph version of 2019/07/06 (https://zenodo.org/record/3267438, accessed on 22 February 2022). The over/under represented GO were selected with a hypergeometric test (FWER value ≤ 0.05). KEGG pathway enrichment analysis was realized with the GAGE R package (V2.40.2) [55], and pathways were selected by Stouffer method with the *q*-value <= 0.05. A map viewer was used for the visualization of DEG associated to pathways [75]. 

The results of the RNAseq differential analysis were confirmed by following the expression level of a selection of 10 genes in leaves of *T. absoluta*-infested and control tomato. The list of targeted genes and primers used is given in Appendix A. The expression level was followed by RT-qPCR. For that purpose, a total of 200 ng of DNA-free RNA samples was reverse transcribed in a final volume of 20 µL using the RNA to cDNA kit (Life technologies, Tokyo, Japan). cDNAs were then diluted 10 times. Gene expression was measured by qPCR on a StepOnePlus Realtime PCR System (ThermoFisher Scientific, Bremen, Germany). Each PCR reaction was made in 20 µL containing primer pairs at 0.3 µM, 40 ng of cDNAs in 1x SYBR^®^ Premix Ex Taq^TM^ (Takara, Shiga, Japan) and followed the amplification protocol consisting in a denaturing step at 95 °C for 30 s, 40 cycles at 95 °C for 5 s and 60 °C for 30 s. The gene expression of each sample was calculated with three analytical replicates and normalized to the two internal control genes *CAC* and *TIP41* [76].

### 4.9. Expression and Purification of Recombinant Proteins

The two full-length sequences (*Solyc11g071470* and *Solyc11g071480*) were cloned from tomato genomic DNA using the primers listed in Appendix A. The PCR products were sub-cloned first in the pCR8 vector (Invitrogen, Carlsbad, CA, USA), prior to sequencing and then transferred into pET28b His-tag expression vector as a *BamH*I-*Not*I fragment. The four expression vectors were introduced into *E. coli* strain BL21-DE3 by heat shock transformation, and the transformants were selected on kanamycin. Protein expression was realized by growing 50 mL of culture at 37 °C until a DO_600_ comprised between 0.4 and 0.6. Then 1 mM of IPTG was added, and the culture was incubated for 22 h at 18 °C. The bacteria were then harvested by a 5000× *g* centrifugation for 15 min at 4 °C, then washed three times in 5 mL of PBS buffer (8.1 mM Na_2_HPO_4_, 1.76 mM KH_2_PO_4_, 2.7 mM KCl, 137 mM NaCl, pH 7.4) and finally resuspended in 1 mL of PBS. Bacterial lysis was realized by three 30 s runs of sonication (Bandelin Sonoplus HD 2010 MS73 probe with an intensity of 200 W/cm²). After a final centrifugation at 16,000× *g* for 10 min at 4 °C, the supernatant containing the protein of interest was collected and stored at −20 °C. His-tag proteins were purified using a Ni-NTA purification system (Qiagen) by following the specification of the manufacturer. Purified His-tag proteins were then used directly to perform enzymatic assays.

### 4.10. Enzymatic Characterization

CoA esters of *p*-coumaric, caffeic, dihydrocaffeic and ferulic acids were synthesized using the recombinant 4-CL enzyme from N. tabacum [77]. Enzymatic assays were performed in a total volume of 100 µL containing potential acyl donors (caffeoyl-, dihydrocaffeoyl-, feruloyl-, and *p*-coumaroyl-CoA) and polyamines (putrescine, cadaverine, agmatine, spermidine, spermine, tyramine, octopamine, dopamine, and noradrenaline) as acyl acceptors in 100 mM Tris-HCl buffer, pH 9, containing 5 mM EDTA. For testing enzyme activity, the reactions were initiated by the addition of purified enzymes, incubated at 35 °C for 5 min, and stopped by adding 50 µL of acetonitrile containing 1% HCl. The reaction mix was then centrifuged 10 min at 14,000× *g* and filtrated on 0.2 µm filter before being analyzed on UPLC-DAD-ESI-MS following the protocol described in [46]. Products were characterized according to their mass and mass fragmentation spectra. Product accumulation was quantified by measuring the area under peak and converted with respect to caffeic, ferulic and *p*-coumaric acid standard curves.

## 5. Conclusions

The present study aimed at characterizing the response of young tomato plants to the herbivory of *T. absoluta* larvae. We developed an integrative approach combining untargeted metabolomic and transcriptomic analyses to characterize both the local and systemic metabolic responses. Our results highlighted an intense local response of the plant to the herbivory of the larvae. In the opposite, the systemic response was far less intense. Regarding the local response, metabolomics analyses revealed a very important increase in the accumulation of a wide range of phenolamides, some of which are reported for the first time in tomato. The transcriptomic analyses indicated a coherent up-regulation of both the polyamine and the phenylpropanoid pathways from which phenolamides are synthesized. In addition, our analyses allowed identifying two genes that we functionally characterized as putrescine hydroxycinnamoyl transferase. Overall, this study brings new comprehensive elements regarding the tomato/*T. absoluta* interaction, which could be used in the development of new strategies to control this pest.

## Figures and Tables

**Figure 1 metabolites-12-00484-f001:**
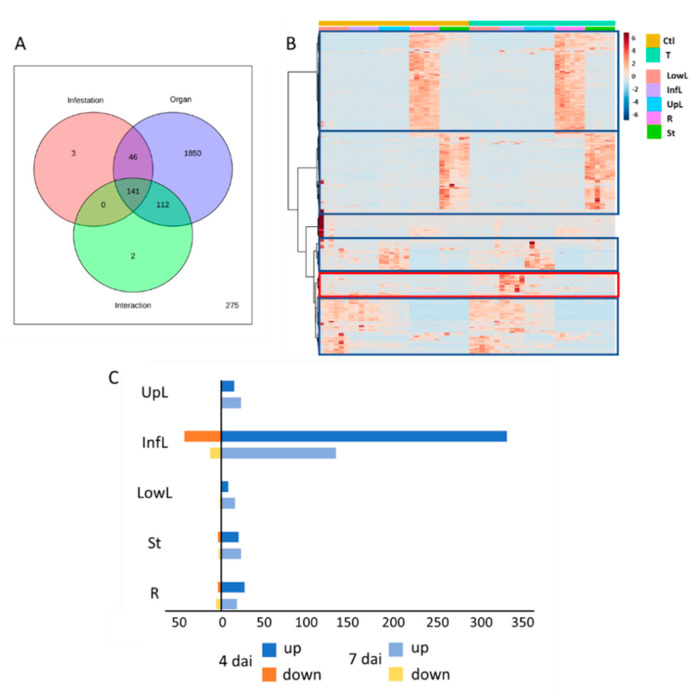
**Impact of the *T. absoluta* herbivory on the metabolome of tomato vegetative organs.** (**A**) Venn diagram representing the number of features significantly affected by the organ nature, *T. absoluta* infestation and their interaction after 4 days of herbivory, determined by a two-way ANOVA (*p* < 0.05). (**B**) Heatmap plot and hierarchical clustering based on the feature intensities from the five vegetative parts of tomato plant submitted or not to *T. absoluta* herbivory for 4 days. Blue squares represent different clusters related to the organ nature. The red square represents the cluster of features affected by *T. absoluta* herbivory. (**C**) Number of differentially accumulated features per organ and per duration of herbivory by considering a log2 Fold Change > 1 and *p* < 0.05 (one-way ANOVA Tukey post-hoc test). The abbreviations for the organ are the following: Infested leaves (InfL), upper leaves (UpL), lower leaves (LowL), stem (St) and roots (R).

**Figure 2 metabolites-12-00484-f002:**
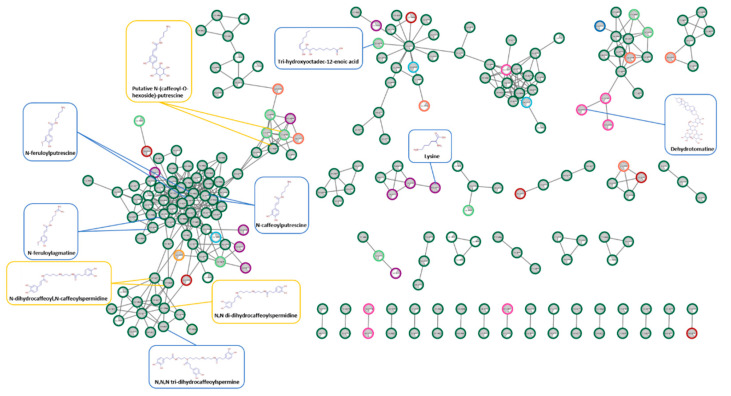
Molecular network of up- and down-regulated features in the tomato organs in response to *T. absoluta* herbivory. The molecular network was obtained using Compound Discoverer 3.3 and was visualized with Cytoscape 3.9.1 following the procedure described at https://github.com/Florent-1/Molecular-Networks-from-Compound-Discoverer-to-Cytoscape (accessed on 10 April 2022). The pie chart in each node represent the relative abundance of the feature in *T.absoluta* treated (grey) vs. control (white). These abundances are presented in priority in InfLat 4 days of herbivory, identified by a dark green circle surrounding the node. When the feature is not affected in InfL, the relative abundance is represented for the most impacted organ, with the following nomenclature: Light green, InfL at 7 days; purple: St at 4 days; pink, St at 7 days; dark blue, LowL at 4 days; light blue, LowL at 7 days; red, R at 4 days; orange, R at 7 days. The structures of some formally identified nodes are surrounded in blue, and those of putative compounds are surrounded in yellow.

**Figure 3 metabolites-12-00484-f003:**
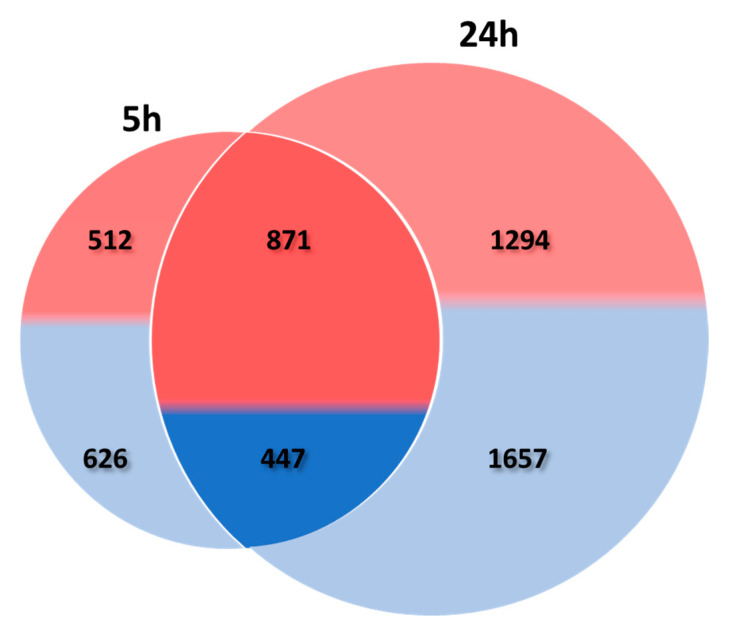
**Venn diagram representing the DEGs in tomato leaf after 5 h and 24 h herbivory.** The proportion of up- and down-regulated DEGs is shown by the proportion of pink/red and blue respectively. The number of up- and down-regulated DEGs is also indicated. The left circle represents DEGs after 5 h, the right circle, DEGs after 24 h, and the intersection between the two circles the DEGs common to 5 h and 24 h of herbivory.

**Figure 4 metabolites-12-00484-f004:**
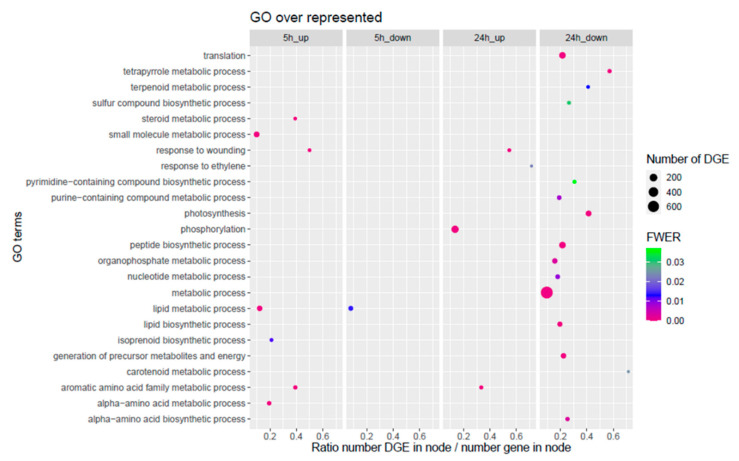
**Gene ontology (GO) terms enrichment at *p* < 0.05 for up- and down-regulated genes after 5 h and 24 h of herbivory.** The size of the circles represents the number of differentially expressed genes (DEG) in each GO category. The position of the circles represents the ratio between the number of DEGs in the GO category to the total number of genes associated to the GO category. The color of the circles represents the statistical significance.

**Figure 5 metabolites-12-00484-f005:**
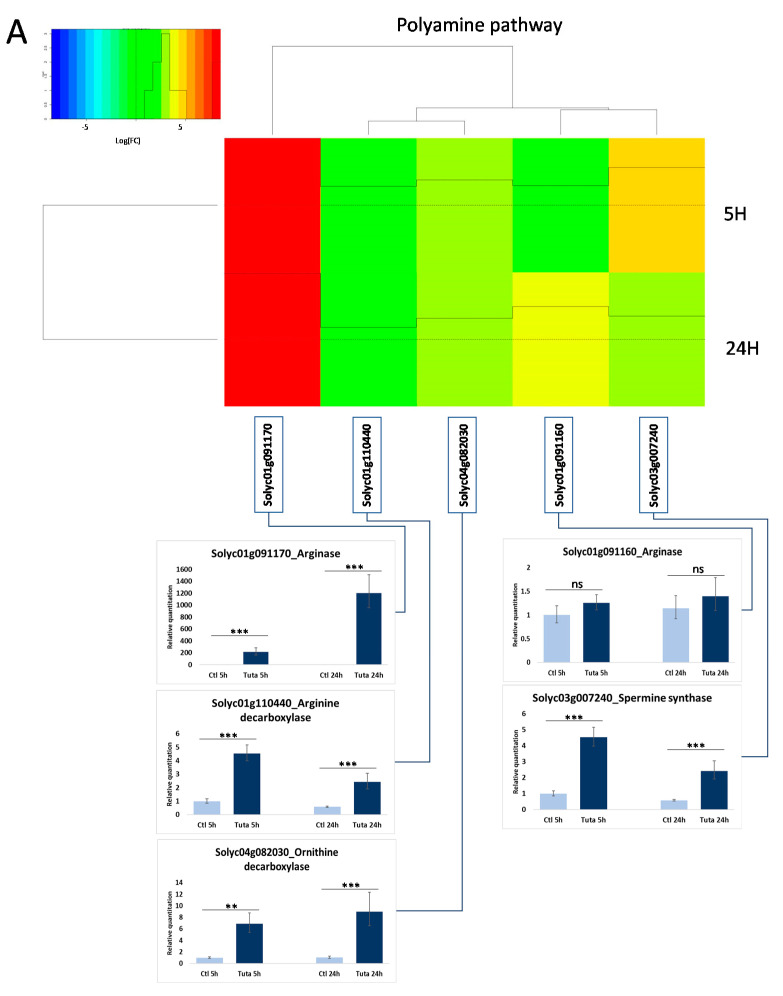
**Expression level of selected genes representing key steps in the polyamine** (**A**) **and phenylpropanoid** (**B**) **pathways.** The heatmap represents the differential expression measured from the RNAseq analysis, between control and *T. absoluta* infested leaf at 5 h and 24 h. In addition, the expression level of several genes was measured by real-time PCR on cDNA prepared from leaves corresponding to the 4 conditions (control and *T. absoluta* infested after 5 h and 24 h). Statistical significance is indicated by * (*p* < 0.05), ** (*p* < 0.01) and *** (*p* < 0.001). ns means non significant.

**Figure 6 metabolites-12-00484-f006:**
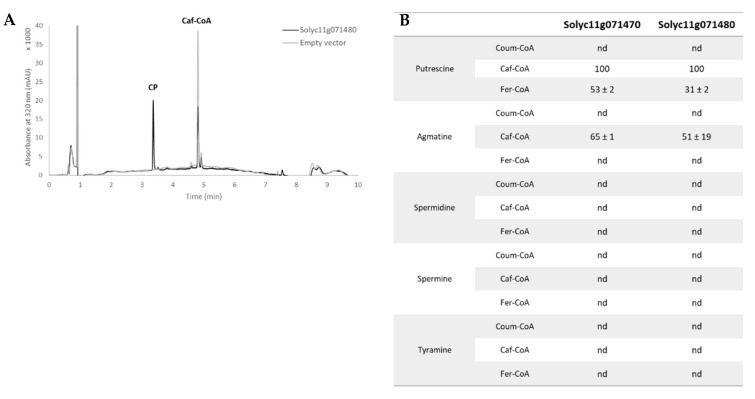
**Functional identification of Solyc11g071470 and Solyc11g071480 as putrescine hydroxycinnamoyl-CoA transferase.** (**A**) Chromatogram of biochemical assays realized on the purified enzymes; (**B**) table showing the substrate preference measured from the consumption of the acyl-CoA substrate and normalized to the best reaction. CP, caffeoyl putrescine; coum-CoA, *p*-coumaroyl-CoA; Caf-CoA, caffeoyl-CoA; Fer-CoA, feruloyl-CoA; nd, no activity detected.

**Figure 7 metabolites-12-00484-f007:**
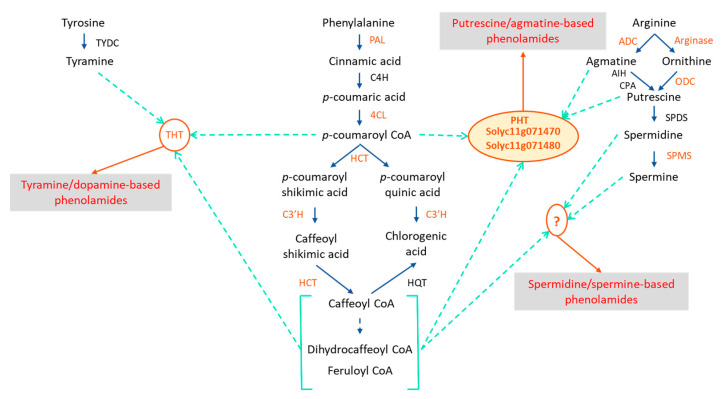
**Schematic representation of the main modulations, at the metabolite and transcript levels, of the phenolamide biosynthesis in response to the herbivory of *T. absoluta***. Genes in orange were shown to be up-regulated upon *T. absoluta* herbivory through RNAseq and/or qPCR. The three phenolamide categories in red were shown to be over-accumulated upon *T. absoluta* herbivory. The blue arrows in solid line represent metabolic steps in the phenylpropanoid and polyamine/monoamine pathways. The light-blue arrows in dashed lines connect the phenylpropanoid and polyamine/monoamine substrates to lead to the synthesis of the three categories of phenolamides catalyzed by three types of enzymes, THT, PHT and a yet unknown enzyme. ADC, arginine decarboxylase; ODC, ornithine decarboxylase; SPDS, spermidine synthase; SPMS, spermine synthase; PAL, phenylalanine ammonia lyase; C4H, cinnamate-4-hydroxylase; 4CL, 4-coumarate CoEnzyme A ligase; HCT, hydroxycinnamoyl quinate/shikimate hydroxycinnamoyl transferase; C3′H, coumaroyl-3′-hydroxylase; HQT, hydroxycinnamoyl quinate hydroxycinnamoyl transferase; TYDC, tyrosine decarboxylase; PHT, putrescine hydroxycinnamoyl transferase; THT, tyramine hydroxycinnamoyl transferase.

**Figure 8 metabolites-12-00484-f008:**
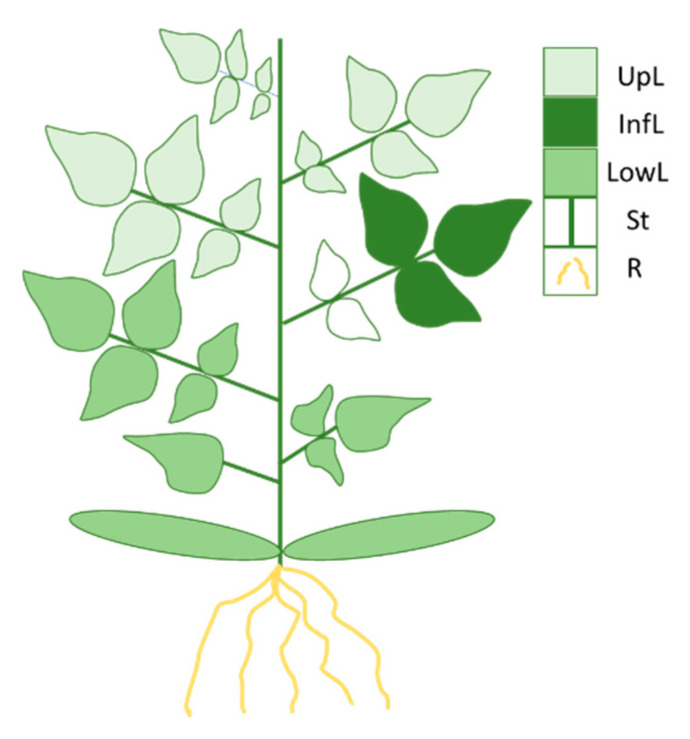
**Scheme of plant parts harvested.** Twelve larvae of *T. absoluta* were deposited on the 3 terminal leaflets of the 4th leaf. After 4 and 7 days of herbivory, these 3 infested leaflets were harvested (InfL). The other plant parts were harvested as indicated: upper leaves (UpL), lower leaves (LowL), stem (St) and roots (R).

**Table 1 metabolites-12-00484-t001:** List of annotated metabolites whose accumulation is impacted by the herbivory of *T. absoluta*. The identification followed the Metabolomics Standard Initiative (MSI) notation [27]. For each metabolite, the foldchange (FC) of the base peak intensity from herbivory treated vs. control was calculated in each organ class and harvest. Statistically significant FC are in bold. Metabolite name followed by an asterisk (*) indicated metabolites presented solely in the *T. absoluta* infested leaf, meaning they could be specific to the presence of the larvae.

Name	Formula	MSI Identification	Metabolite Family	Calc. MW	*m/z*	RT [min]	FC *T. absoluta* Herbivory/Control									
							LowL_4d	InfL_4d	UpL_4d	R_4d	St_4d	LowL_7d	InfL_7d	UpL_7d	R_7d	St_7d
*p*-coumaroyltyramine	C_17_H_17_NO_3_	L2	Phenolamide	283.121	284.12834	18.4	1.2	**11**	0.9	0.7	1.0	1.4	**5.0**	0.7	0.6	1.7
*p*-coumaroylputrescine	C_13_H_18_N_2_O_2_	L1	Phenolamide	234.1369	235.14417	6.8	1.2	**31**	1.9	2.1	1.0	1.7	9.3	1.3	0.7	1.7
*p*-coumaroylputrescine	C_13_H_18_N_2_O_2_	L1	Phenolamide	234.1369	235.14413	7.2	1.3	**17**	1.0	1.0	1.3	0.9	1.8	1.3	0.9	1.6
*p*-coumaroylputrescine	C_13_H_18_N_2_O_2_	L1	Phenolamide	234.137	235.14424	3.8	1.1	**4.8**	1.2	1.3	1.4	1.3	**2.4**	1.6	0.8	1.6
*p*-coumaroyloctopamine	C_17_H_17_NO_4_	L2	Phenolamide	299.1159	300.12324	17.5	1.3	**9.7**	1.2	1.5	0.9	4.1	1.1	1.1	1.3	1.0
*N*-feruloyltyramine	C_18_H_19_NO_4_	L2	Phenolamide	313.1315	314.13881	18.6	1.1	**10**	0.9	0.8	1.2	1.5	**7.0**	0.8	0.9	2.0
*N*-feruloyltyramine	C_18_H_19_NO_4_	L2	Phenolamide	313.1315	314.13878	15.9	1.2	**3.0**	1.4	1.0	1.3	1.3	1.6	1.3	1.0	1.2
*N*-feruloylputrescine	C_14_H_20_N_2_O_3_	L1	Phenolamide	264.1473	265.15461	9.6	1.2	**23**	1.3	1.0	1.2	1.0	4.2	1.3	0.7	0.9
*N*-feruloylputrescine	C_14_H_20_N_2_O_3_	L1	Phenolamide	264.1474	265.1547	5.5	1.4	**13**	1.2	1.0	1.1	1.8	5.9	1.4	0.7	1.2
*N*-feruloyl-*O*-methyl-dehydrodopamine	C_19_H_19_NO_5_	L2	Phenolamide	341.1264	342.13364	17.3	2.7	**12**	1.1	0.7	0.8	3.9	4.2	1.0	1.2	0.9
*N*-feruloyllysine	C_16_H_22_N_2_O_5_	L2	Phenolamide	322.153	323.16024	7.7	1.2	**21**	1.4	1.0	1.3	2.4	**6.1**	2.8	1.4	1.5
*N*-feruloyllysine	C_16_H_22_N_2_O_5_	L2	Phenolamide	322.1529	323.16018	12.2	**3.3**	**8.4**	2.3	1.9	**3.2**	2.4	**16**	1.4	2.5	1.7
*N*-feruloyllysine	C_16_H_22_N_2_O_5_	L2	Phenolamide	322.1529	323.16019	9.2	2.0	**3.0**	1.9	1.7	1.9	2.3	**7.4**	1.4	1.7	2.1
*N*-feruloyl dehydrotyramine	C_18_H_17_NO_4_	L2	Phenolamide	311.1159	312.12321	17.1	0.9	**53**	0.9	0.9	0.8	1.8	**34**	1.1	0.9	1.0
*N*-feruloylagmatine	C_15_H_22_N_4_O_3_	L1	Phenolamide	306.1692	307.17645	12.5	1.2	**4.8**	1.2	1.1	1.1	1.2	1.9	1.0	1.1	1.2
*N*-feruloylagmatine	C_15_H_22_N_4_O_3_	L1	Phenolamide	306.1692	307.17649	9.0	1.1	**4.0**	1.3	1.2	1.0	1.1	1.7	1.2	1.0	1.1
*N*-feruloyl *O*-methyldopamine	C_19_H_21_NO_5_	L2	Phenolamide	343.1421	344.14936	18.8	1.2	**9.0**	1.0	1.1	1.1	2.0	**4.2**	0.9	0.9	1.8
*N*-feruloyl dehydrotyramine	C_18_H_17_NO_4_	L2	Phenolamide	311.1159	312.12318	15.6	1.1	**8.3**	1.0	1.0	1.0	2.1	4.3	0.9	0.9	1.1
*N*-dihydroferuloylputrescine	C_14_H_22_N_2_O_3_	L1	Phenolamide	266.1631	267.17036	5.2	1.4	**19**	1.5	1.5	1.6	1.2	**9.8**	1.5	1.0	1.6
*N*-dihydrocaffeoylspermine	C_19_H_34_N_4_O_3_	L2	Phenolamide	366.263	367.27031	7.6	1.5	**7.3**	1.5	1.2	2.4	1.4	2.9	1.8	1.2	1.8
*N*-dihydrocaffeoylputrescine	C_13_H_20_N_2_O_3_	L1	Phenolamide	252.1474	253.15468	2.7	2.2	**90**	2.5	2.4	2.0	1.4	**24**	2.8	1.3	1.6
*N*-dihydrocaffeoyl, *N*-caffeoylspermidine	C_25_H_33_N_3_O_6_	L2	Phenolamide	471.2369	472.24416	13.0	2.3	**20**	**3.8**	1.1	2.4	1.8	**9.4**	**2.8**	1.5	2.2
*p*-coumaroylagmatine	C_14_H_20_N_4_O_2_	L1	Phenolamide	276.1588	277.16603	11.1	1.1	**10**	1.5	1.4	2.0	1.4	2.6	1.4	0.9	1.7
*p*-coumaroylagmatine	C_14_H_20_N_4_O_2_	L1	Phenolamide	276.1587	277.166	7.1	1.2	**6.4**	1.3	2.6	1.4	0.9	3.5	1.0	0.5	1.2
*N*-*cis*-(dihydrocaffeoyl-*O*-hexoside)-putrescine	C_19_H_30_N_2_O_8_	L2	Phenolamide	414.2003	415.20757	2.7	1.2	1.6	1.3	0.8	1.7	0.6	**4.5**	4.2	2.0	1.4
*N*-*cis*-(Caffeoyl-*O*-hexoside)-putrescine	C_19_H_28_N_2_O_8_	L2	Phenolamide	412.1844	413.19171	4.5	1.6	**14**	0.9	1.4	1.4	1.0	**11**	1.9	0.9	1.1
*N*-*cis*-(caffeoyl-*O*-hexoside)-putrescine	C_19_H_28_N_2_O_8_	L2	Phenolamide	412.1846	413.19192	3.5	0.9	**2.7**	0.8	1.5	1.2	1.3	2.2	1.0	1.6	1.4
*N*-*cis*-(caffeoyl-*O*-hexoside)-putrescine	C_19_H_28_N_2_O_8_	L2	Phenolamide	412.1846	413.19188	2.8	1.2	2.0	1.0	1.3	1.0	1.2	**5.5**	2.2	0.9	1.4
*N*-cinnamoylputrescine	C_13_H_18_N_2_O	L2	Phenolamide	218.142	219.14926	12.3	1.3	**8.3**	1.3	0.9	1.3	1.6	**8.0**	1.8	1.0	1.3
*N*-caffeoylputrescine	C_13_H_18_N_2_O_3_	L1	Phenolamide	250.1316	249.12424	4.3	1.4	**29**	1.5	1.3	1.3	1.0	**17**	1.3	0.9	1.2
*N*-caffeoylputrescine	C_13_H_18_N_2_O_3_	L1	Phenolamide	250.1317	251.139	2.7	1.2	**7.1**	1.3	1.0	1.1	1.5	**3.3**	1.6	1.0	1.6
*N*-caffeoyllysine	C_15_H_20_N_2_O_5_	L2	Phenolamide	308.1372	309.14447	8.0	**4.2**	**8.2**	4.7	1.0	**7.8**	3.7	**14**	1.8	0.9	3.2
*N*-caffeoylagmatine	C_14_H_20_N_4_O_3_	L1	Phenolamide	292.1537	293.16097	9.2	1.2	**12**	1.8	1.8	1.3	1.2	3.9	1.7	1.1	1.5
*N*-caffeoyl,*N*-dihydrocaffeoylspermidine	C_25_H_33_N_3_O_6_	L2	Phenolamide	471.2369	472.24418	13.5	2.3	**20**	2.5	0.9	2.2	1.0	**13**	3.2	2.4	2.1
*N*,*N*-dihydrocaffeoylspermidine	C_25_H_35_N_3_O_6_	L2	Phenolamide	473.2524	474.25965	12.0	**3.2**	**48**	4.3	3.0	2.6	2.1	**11**	4.3	2.4	1.7
*N*,*N*-*bis*-dihydrocaffeoyl,*N*-caffeoylspermine	C_37_H_48_N_4_O_9_	L2	Phenolamide	692.3417	693.34895	14.9	1.2	**16**	2.1	1.0	2.4	2.2	**13**	**13**	0.8	3.4
*N*,*N*,*N tri*-dihydrocaffeoylspermine	C_37_H_50_N_4_O_9_	L1	Phenolamide	694.357	695.36422	15.8	2.1	**17**	3.4	2.8	2.0	2.0	7.1	3.1	2.1	1.7
*N*, *N bis*-dihydrocaffeoyl, *N*-caffeoylspermine	C_37_H_48_N_4_O_9_	L2	Phenolamide	692.3419	693.34916	16.4	2.9	**48**	**5.2**	1.1	3.7	1.6	**20**	**6.8**	2.5	**5.5**
*N*, *N bis*-dihydrocaffeoyl, *N*-caffeoylspermine	C_37_H_48_N_4_O_9_	L2	Phenolamide	692.3417	693.34899	16.0	1.6	**11**	3.3	1.6	1.6	1.1	**5.5**	3.1	2.3	2.6
*N*, *N bis*-caffeoyl, *N*-dihydrocaffeoylspermine	C_37_H_46_N_4_O_9_	L2	Phenolamide	690.3263	691.33353	16.5	2.7	**11**	**3.5**	1.1	3.1	1.2	**18**	4.5	1.4	1.7
caffeoyl-lysine glucoside	C_21_H_30_N_2_O_10_	L2	Phenolamide	470.1898	471.19706	8.6	0.8	1.2	1.3	1.6	**8.4**	0.7	1.1	5.0	1.4	4.8
3-(4-hydroxyphenyl)-*N*-[(1*E*)-2-(4-hydroxyphenyl)ethenyl]-(2*E*)-2-propenamide	C_17_H_15_NO_3_	L2	Phenolamide	281.1055	282.11278	16.6	1.5	**88**	1.2	1.1	0.7	1.1	**28**	1.0	0.7	1.0
(2*E*)-*N*-(4-acetamidobutyl)-3-(4-hydroxy-3-methoxyphenyl)prop-2-enamide	C_16_H_22_N_2_O_4_	L3	Phenolamide	306.1581	307.16533	13.3	1.0	0.8	2.7	1.1	**5.2**	1.2	0.8	3.5	0.9	3.8
Putative feruloyl-*N*_6_-methylagmatine	C_16_H_24_N_4_O_3_	L2	Phenolamide	320.1849	321.19217	13.3	1.3	**5.1**	1.2	1.1	1.2	1.3	2.3	1.2	1.1	1.2
Putative feruloyl-*N*_6_-methylagmatine	C_16_H_24_N_4_O_3_	L2	Phenolamide	320.1849	321.19213	10.3	1.4	**3.0**	1.3	1.2	0.8	1.3	1.5	1.0	1.1	1.1
Methyl-(10*R*)-hydroxy-(11*S*,12*S*)-epoxy-(5*Z*,8*Z*,14*Z*)-eicosatrienoate	C_21_H_34_O_4_	L3	Lipid	350.2458	351.25302	22.9	0.8	1.8	0.8	**0.4**	1.0	0.9	1.2	1.3	2.8	1.3
hydroxyprogesterone caproate	C_27_H_40_O_4_	L3	Lipid	428.2906	429.29786	23.8	0.8	0.6	1.2	1.0	0.7	**2.9**	1.0	0.4	0.9	1.0
[FA(18:4)]6*Z*_9*Z*_12*Z*_15*Z*-octadecatetraenoicacid	C_18_H_28_O_2_	L3	Lipid	276.2092	277.21647	23.8	0.7	**3.8**	0.7	1.0	0.8	0.9	1.4	0.8	0.8	1.1
[FA(18:3)]13S-hydroperoxy-9*Z*_11*E*_14*Z*-octadecatrienoicacid	C_18_H_30_O_4_	L3	Lipid	310.2144	309.20711	22.0	0.9	**14**	0.6	1.1	1.1	0.8	**4.7**	1.5	0.9	0.8
[FA(18:3)]13S-hydroperoxy-9*Z*_11*E*_14*Z*-octadecatrienoicacid	C_18_H_30_O_4_	L3	Lipid	310.2144	309.20703	22.2	1.1	**5.3**	0.8	1.0	1.1	0.9	**3.8**	0.9	1.0	1.2
(12Z)-9,10,11-trihydroxyoctadec-12-enoic acid	C_18_H_34_O_5_	L3	Lipid	352.2228	353.23006	21.8	0.7	4.7	2.0	1.2	1.1	1.5	**6.5**	1.1	1.7	0.9
Solasodin	C_27_H_43_NO_2_	L2	Glycoalkaloid	413.3293	414.33655	19.4	1.0	0.9	1.0	1.0	1.3	1.2	1.0	1.1	1.0	**2.0**
Solasodin	C_27_H_43_NO_2_	L2	Glycoalkaloid	413.3296	414.33683	20.9	1.0	2.3	1.2	0.7	1.1	1.8	2.2	1.2	0.8	**2.3**
Dehydrotomatine	C_50_H_81_NO_21_	L2	Glycoalkaloid	2063.058	1032.5365	19.4	1.3	1.1	1.2	1.0	1.5	1.3	1.2	1.5	0.9	**2.5**
Gossypetin 7,4′-dimethyl ether 8-acetate	C_19_H_16_O_9_	L3	Flavonoïd	388.0796	389.08683	15.5	1.1	**0.3**	0.7	1.0	1.0	1.6	1.3	1.6	1.1	1.2
Proclavaminic acid *	C_8_H_14_N_2_O_4_	L3	Carboxylic acid	202.0953	203.10259	2.4	1.0	**101**	1.0	2.4	0.5	1.3	**33**	1.0	1.6	0.9
Decylubiquinone	C_19_H_30_O_4_	L3	Benzoquinone	322.2146	323.22185	21.8	0.9	**5.4**	0.5	0.9	1.0	1.3	2.3	3.0	1.2	1.0
Decylubiquinone	C_19_H_30_O_4_	L3	Benzoquinone	322.2145	323.22182	22.0	1.1	**6.4**	0.7	0.8	1.2	1.3	2.6	1.7	0.3	1.3
Guanine	C_5_H_5_N_5_O	L2	Amino purine	151.0494	152.0567	3.9	0.9	**4.4**	1.0	0.7	0.8	0.6	1.5	1.0	0.9	1.3
Phenyl-butyryl-glutamine	C_15_H_20_N_2_O_4_	L2	Amino acid	292.1424	293.14972	6.8	1.4	4.0	4.6	0.8	**6.0**	5.5	**13**	2.8	0.7	4.4
Phenyl-butyryl-glutamine	C_15_H_20_N_2_O_4_	L2	Amino acid	292.1425	293.14974	10.4	**7.4**	**6.2**	**7.2**	0.9	**9.5**	**4.5**	7.5	3.7	0.9	4.1
*N*-decanoyl histidine	C_16_H_27_N_3_O_3_	L2	Amino acid	309.2054	310.21263	7.6	1.6	**5.4**	1.9	1.2	2.4	1.7	2.2	1.5	1.1	1.4
*N*-acetyltyramine	C_10_H_13_NO_2_	L3	Amino acid	179.0946	180.1019	7.0	1.0	**2.0**	**2.8**	1.0	1.0	1.7	1.9	1.3	0.5	1.0
*N*_6__*N*_6__*N*_6_-Trimethyl-L-lysine	C_9_H_20_N_2_O	L2	Amino acid	172.1576	173.16488	3.2	1.3	**2.4**	1.0	0.9	1.2	1.2	**2.9**	1.0	0.8	1.3
*N*_6__*N*_6__*N*_6_-Trimethyl-L-lysine	C_9_H_20_N_2_O	L3	Amino acid	172.1576	173.16488	1.9	1.0	**77**	1.3	0.9	1.7	1.5	**31**	0.9	1.1	1.2
Malonyltryptophan	C_14_H_14_N_2_O_5_	L1	Amino acid	290.0904	291.09769	15.8	1.3	**2.9**	1.6	1.0	2.1	1.9	**8.2**	1.4	0.9	2.5
Lysine	C_6_H_14_N_2_O_2_	L1	Amino acid	146.1055	147.11281	8.0	1.6	1.5	3.2	1.0	**7.1**	3.2	1.8	1.7	0.8	3.5
Capryloylglycine	C_10_H_19_NO_3_	L3	Amino acid	201.1365	202.1438	4.2	1.1	0.7	1.0	0.5	1.8	1.1	0.8	1.3	**0.1**	1.0
Butyryl-carnitine *	C_11_H_21_NO_4_	L3	Amino acid	231.1471	232.15434	3.1	0.8	**98**	1.0	1.2	1.3	0.7	**52**	2.5	2.1	1.5
9-amino-nonanoic acid	C_9_H_19_NO_2_	L3	Amino acid	173.1416	174.14892	10.1	1.0	**2.4**	1.3	1.0	1.3	1.4	2.0	0.9	1.0	1.6
2-Acetamido-2-deoxy-3-*O*-beta-D-galactopyranosyl-1-*O*-*L*-threonyl-alpha-*D*-galactopyranose	C_18_H_32_N_2_O_13_	L3	Unknown	484.1923	485.19958	17.7	0.9	**5.0**	1.6	1.4	0.8	0.8	2.0	1.1	1.0	1.2
19-Nortestosterone	C_18_H_26_O_2_	L3	Terpenoid	274.1935	275.2008	22.0	0.7	**8.1**	1.1	1.0	1.2	0.8	2.4	1.3	1.2	0.7
5′-S-Methyl-5′-thioadenosine	C_11_H_15_N_5_O_3_S	L2	Ribonucleoside	297.0896	298.09688	8.3	1.3	**0.1**	1.2	1.0	0.9	2.0	0.8	0.4	0.8	1.0
O-feruloylquinate	C_17_H_20_O_9_	L2	Phenolic acid	368.1107	369.11793	12.5	3.2	1.1	1.0	0.9	1.1	1.2	**0.2**	1.1	1.1	1.5
Ferulic acid	C_10_H_10_O_4_	L1	Phenolic acid	194.058	195.06521	12.6	1.3	**0.5**	0.8	1.1	0.8	1.1	1.0	0.8	0.9	0.9
1-*O*-feruloyl-beta-*D*-glucose	C_16_H_20_O_9_	L2	Phenolic acid	356.1107	355.1032	11.3	2.0	**6.7**	2.2	0.9	1.6	0.9	2.0	2.0	1.0	1.2

## Data Availability

The RNA libraries are available on Gene Expression Omnibus (GEO) under the ID GSE200795. The sequences of *Solyc11g071470* and *Solyc11g071480* are available on Genbank ON248950 and ON248951 respectively.

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
