# Peer review of "Transcriptomics and Metabolomics Analyses Reveal High Induction of the Phenolamide Pathway in Tomato Plants Attacked by the Leafminer Tuta absoluta"

_metabolites, 2022, doi:10.3390/metabo12060484_

Round 1

Reviewer 1 Report

1. Why the leaf positions for metabolomic analysis (4th true leaf) and RNA-seq (3rd true leaf) are different?

2. At what time when the treatments start? Because the plant response can be affected by the time in a day.

3. I am worried if the mixture of many leaves for the upper leaves and lower leaves may affect the results. Since each leaves has a different response to indirect herbivory signals.

Author Response

We thank reviewer 1 comments and give answers below:

Reviewer 1: Why the leaf positions for metabolomic analysis (4th true leaf) and RNA-seq (3rd true leaf) are different?

Authors : Thanks to the reviewer for the analysis and for raising this error.The larvae were deposited on the same leaf position for the metabomics and RNAseq experiements. Now this is corrected.

Reviewer 1: At what time when the treatments start? Because the plant response can be affected by the time in a day.

Authors : For each experiment, the larvae were deposited in the morning between 10 and 11 am. Even if the plant response could be affected by the moment in the day, in this case, the herbivory represents a continuous stress that lasts for several hours to several days. For this reason we do not believe it could have an impact on the results presented and decide not to precise this information in the text.

Reviewer 1: I am worried if the mixture of many leaves for the upper leaves and lower leaves may affect the results. Since each leaves has a different response to indirect herbivory signals.

Authors : We agree with the reviewer regarding the point that the individual response of each leaf may be different, however the idea here was more to show that the reponse to herbivory is detectable outside the infested leaf (constitutive response) than to describe the individual response of each leaf.

Reviewer 2 Report

Comments for authors
This paper entitled “Transcriptomics and metabolomics analyses reveal high induction of the phenolamide pathway in tomato plants attacked by the leafminer Tuta absoluta” by Roumani et al. reported metabolic responses of young tomato plants to T. absoluta larvae herbivory, especially for the accumulation of phenolamides, by integrating untargeted metabolomic and transcriptomic analyses. In addition, the authors characterized putrescine hydroxycinnamoyl transferases which might be involved in the biosynthesis of several phenolamides. Furthermore, the experimental designs are appropriate and the data presented look reliable. In my opinion, the manuscript should be suitable for publication in the Metabolites after a few minor changes, as below indicated:
Some minor points.
1.    Lines 70 and 73, “Z” should be Italic.
2.    Line 95, “T. absoluta” should be Italic.
3.    Line 148, “methanol 70%” should be changed into “70% aqueous methanol”.
4.    Line 152, “Vanquish” should be changed into “Vanquish UHPLC system”.
5.    Line 154, “150*2.1” should be changed into “150*2.1 mm”.
6.    Line 156, “MetOH” should be changed into “methanol”.
7.    Line 166, “120 to 1200” should be changed into “120 to 1200 m/z”.
8.    Line 174, “U-HPLC-HRMS” should be changed into “UHPLC-HRMS”.
9.    Line 189, “HR-LC/MS” should be changed into “LC-HRMS”.
10.    The authors should add a description of the LC-DAD-MS detection method used in the enzymatic characterization.
11.    The Supplementary Materials are missing.

Author Response

We thanks reviewer 2 for their comments and give answers below:

Reviewer 2: Lines 70 and 73, “Z” should be Italic.

Authors :These two characters are already Italic.
Reviewer 2: Line 95, “T. absoluta” should be Italic.

Authors : This term is already Italic
Reviewer 2: Line 148, “methanol 70%” should be changed into “70% aqueous methanol”. Authors :  Done
Reviewer 2: Line 152, “Vanquish” should be changed into “Vanquish UHPLC system”. Authors : Done
Reviewer 2: Line 154, “150*2.1” should be changed into “150*2.1 mm”.

Authors : Done
Reviewer 2: Line 156, “MetOH” should be changed into “methanol”.

Authors :Done
Reviewer 2: Line 166, “120 to 1200” should be changed into “120 to 1200 m/z”.

Authors : Done
Reviewer 2: Line 174, “U-HPLC-HRMS” should be changed into “UHPLC-HRMS”. Authors :Done
Reviewer 2: Line 189, “HR-LC/MS” should be changed into “LC-HRMS”.

Authors :Done
Reviewer 2: The authors should add a description of the LC-DAD-MS detection method used in the enzymatic characterization.

Authors :We have referred to a previous article describing the LC-DAD-MS detection method (Larbat et al., 2014)
Reviewer 2: The Supplementary Materials are missing.

Authors :The Supplementary Materials were deposited has several Word and Excel files on the Metabolites website. Please check with the journal administrators to fix this issue.

Reviewer 3 Report

This is a good manuscript, the subject matter is important. The results are interesting, as a response mechanism of a plant to the attack of herbivores. The authors point out the accumulation of phenolamide-type compounds, which could be responsible for the plant's defensive response. It would be interesting to include a possible mechanism of action in the discussion of the results. Would it be possible that these compounds could act as agonists or antagonists with some of the pathways of the insect's nervous system?

Author Response

We thank reviewer 3 for their comments and give answers below:

Reviewer 3 : It would be interesting to include a possible mechanism of action in the discussion of the results. Would it be possible that these compounds could act as agonists or antagonists with some of the pathways of the insect's nervous system?

Author : We thank the reviewer for this analysis and questions. Unfortunately, the actual knowledge is close to zero regarding the possible mode of action of phenolamides on insect toxicity. We are currently involved in a collaboration dedicated to study this aspect, but so far mentionning any element in the discussion part would be highly hypothetical.